# Holocene Sea-Level Changes in Southern Brazil Based on High-Resolution Radar Stratigraphy



**Eduardo Guimarães Barboza** [1,2,*] **, Sergio Rebello Dillenburg** [1] **, Matias do Nascimento Ritter** [2] **,
Rodolfo José Angulo** [3] **, Anderson Biancini da Silva** [4] **, Maria Luiza Correaa da Camara Rosa** [1] **, Felipe Caron** [2]
**and Maria Cristina de Souza** [3]

1    Centro de Estudos de Geologia Costeira e Oceânica, Instituto de Geociências, Universidade Federal do Rio
     Grande do Sul, Porto Alegre 91509-900, RS, Brazil; sergio.dillenburg@ufrgs.br (S.R.D.);
     luiza.camara@ufrgs.br (M.L.C.d.C.R.)
2    Centro de Estudos Costeiros, Limnológicos e Marinhos, Campus do Litoral, Universidade Federal do Rio
     Grande do Sul, Imbé 95625-000, RS, Brazil; matias.ritter@ufrgs.br (M.d.N.R.); felipe.caron@ufrgs.br (F.C.)
3    Laboratório de Estudos Costeiros, Departamento de Geologia, Universidade Federal do Paraná,
     Curitiba 81530-000, PR, Brazil; fitoangulo@gmail.com (R.J.A.); cristinasouza2527@gmail.com (M.C.d.S.)
4    Gerência de Licenciamento Ambiental de Atividades Estratégicas, Instituto do Meio Ambiente de Santa
     Catarina, Florianópolis 88020-300, SC, Brazil; andersondasilva@ima.sc.gov.br
*    Correspondence: eduardo.barboza@ufrgs.br; Tel.: +55-51984776200

**Abstract:** This paper focuses on high-resolution coastal stratigraphy data, which were revealed by the Ground Penetrating Radar (GPR) system. Surveys performed with GPR on the surface of prograded barriers reveal patterns of reflections that allow the interpretation of the geometry and stratigraphy of coastal deposits in a continuous mode. At the Curumim prograded barrier in southern Brazil (29°30′ S–49°53′ W), a two-dimensional transverse GPR survey revealed, with high precision, a serial of contacts between aeolian deposits of relict foredunes and relict beach deposits that have a strong correlation with sea level. In a 4 km GPR profile, a total of 24 of these contacts were identified. The high accurate spatial positioning of the contacts combined with Optical Stimulated Luminescence dating resulted in the first confident sea-level curve that tells the history of sea-level changes during the last 7 ka on the southernmost sector of the Brazilian coast. The curve shows that sea-level was still rising before 6 ka BP, with a maximum level of 1.9 m reached close to 5 ka BP; after that, sea-level started to falling slowly until around 4 ka BP when fall accelerated.

**Keywords:** sea-level curve; coastal evolution; prograded barrier; GPR

## 1. Introduction

The pursuit of a confident sea-level curve representing oscillations of the sea level during the middle and late Holocene has been a constant goal in the last decades. In Brazil, since 1979, when the first most detailed curve was proposed [1], and later contested [2], two groups have debated the existence of high-frequency oscillations (2–3 m), operating in the scale of centuries (500–600 years), that could have occurred during the overall sea-level fall established after a maximum level of few meters was reached between 6–5 ka BP [3,4]. Since then, high-frequency oscillations have lost credibility. Moreover, one reason for that is that strong (convincing) dated coastal geomorphological records were never formally presented, confirming such oscillations. From all indicators of paleo sea-levels used by those authors, such as marine (beach), lagoon and mangrove deposits, vermetid incrustations and coral reefs, and even archaeological indicators such as "Sambaquis", the vermetid incrustations got more credibility and confidence as a sea-level indicator [4].

In South Brazil, to the south of the Santa Marta Cape (28°35′ S-48°49′ W–Figure 1), a formal and confident curve was never proposed due to the lack of confident indicators of paleo sea level along with this almost entirely sandy coast. For instance, vermetid incrustations occur fixed on coastal rocks, which do not occur to the south of the Santa

Marta Cape, except for Torres (29°20′ S–49°44′ W–Figure 1) beach, where a rock promontory occurs, which has no occurrence of fossil vermetid. A very general curve was proposed but not supported by any of the indicators mentioned above of paleo sea levels [5]. Later, a lagoonal terrace positioned at around +3 m asl was related to the maximum sea level reached between 6–5 ka BP [6], and the contact between aeolian and beach deposits in cores of the prograded Curumim barrier (29°38′ S–Figure 1) was correlated with paleo sea levels [7]. These contacts were also identified in a GPR record of a prograded barrier located 50 km to the north of Curumim barrier at the Passo de Torres (29°15′ S–49°42′ W–Figure 1) and correlated with paleo sea levels [8]. Only recently, in South Brazil, better attention was given to this paleo sea-level indicator (aeolian/beach contact), along the prograded Cassino barrier (32°10′ S-52°11′ W–Figure 1), where a maximum sea-level of 2.0 m reached at 6–5 ka BP and after that followed by an overall sea-level fall was determined [9]. This indicator of paleo sea level was detected in eight cores along a 15 km long transect on the prograded barrier of Cassino. The contact was defined with a precision of 0.5 m due to the remarkably high contrast of compaction between relict aeolian sand deposits of foredunes and backshore sands occurring underneath. The chronology of barrier progradation was determined by a combination of radiocarbon ($^{14}$C) and Optical Stimulated Luminescence (OSL) datings. The limitation at the Cassino barrier was the necessity of a high-cost drilling system and the consequent discontinuity of such records along a transect.

After the pioneer study that combined the dune-beach facies boundary and the dating of the dune base as an indicator of paleo sea-level [10], some new studies have used the relict top of the berm facies [11] or the raised beach strata [12] as indicators of paleo sea levels. More recently, converging methods of paleo sea-level reconstructions were presented [13,14], based on the contact between relict foredunes and backshore deposits along with prograded barriers. The advantage of these methods is the more precise identification of the foredune/backshore deposits contact in GPR records. The continuity of the contact in GPR records makes it possible to determine a great number of spatial positions of such contacts. Combined with OSL datings (informing the chronology of the development of a prograded barrier), the method allows the reconstruction of sea-level behavior during the period of barrier progradation.

In a sandy coastal region as southern Brazil, showing two large prograded barriers (Cassino and Curumim), we choose the Curumim barrier to extract relict records of paleo sea levels. By combining high-quality geophysical records obtained in GPR surveys, OSL dating, and determinations of altitudes by a Global Navigation Satellite System (GNSS) system, the reconstruction of sea-level behavior during 4 km progradation of the Curumim barrier was made possible.



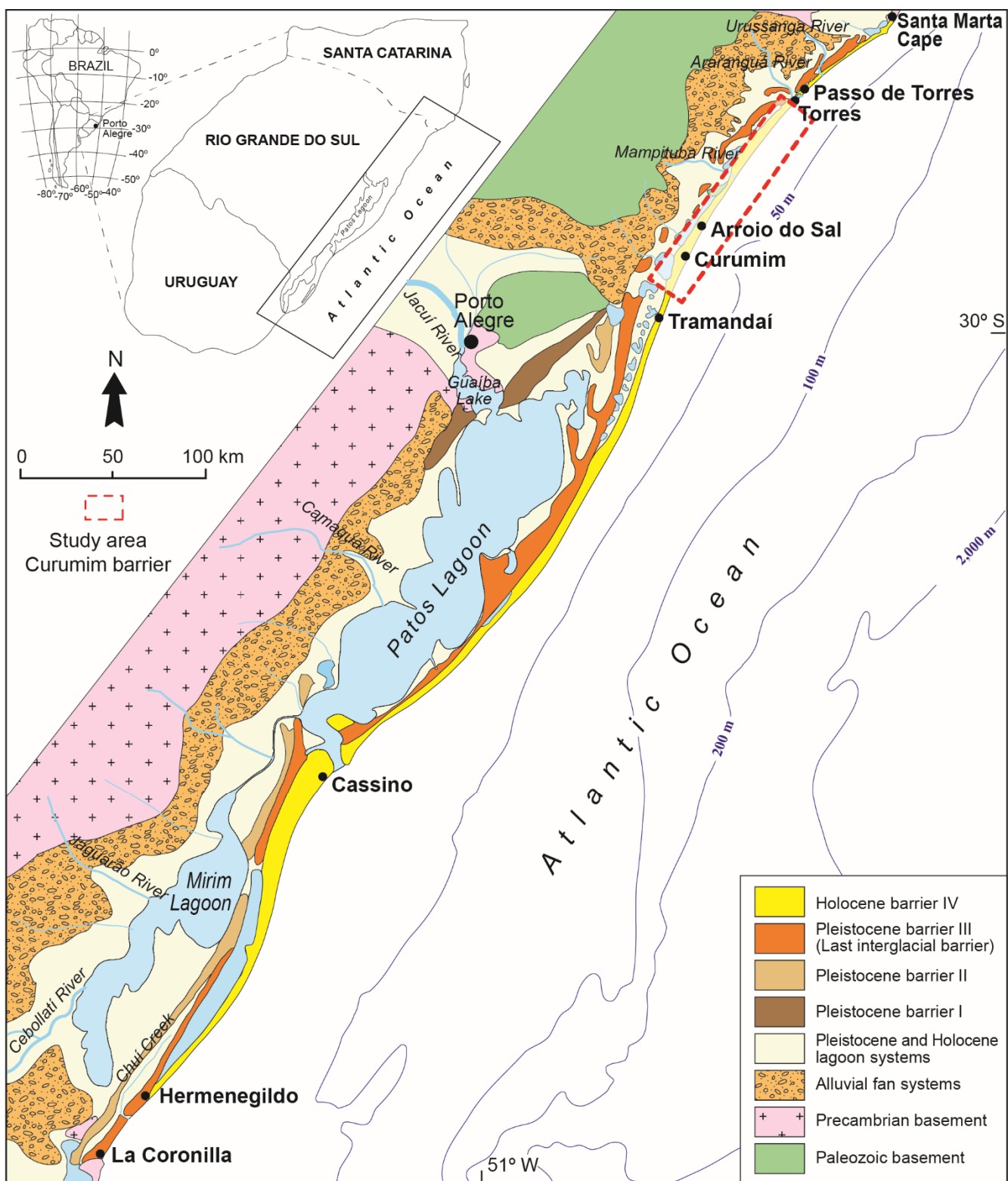

**Figure 1.** Location of the Curumim barrier study area (dashed red rectangle) showing the general geology of the coastal plain of southern Brazil–datum WGS84 (modified after [15]).

## 2. Regional Setting

Curumim gives name to the prograded (or regressive) barrier that was formed in a very gentle embayment between Tramandaí and Torres [15] (Figure 1). The coastal plain where the Curumim barrier occurs is characterized by coastal sandy barriers and lagoons formed during the Quaternary [16], and its location in an intra-plate position suggests a tectonically stable condition [17]. There is no indication that basin subsidence and sediment compaction were relevant during the Holocene [18]. Right at Curumim, the barrier shows its widest dimension (5 km) due to barrier progradation during the last 7.3 ka [19]. The prograded plain of the barrier is completely and dominantly covered by aeolian deposits of transgressive sand sheets (TSS), and rare transgressive dunefields. At least 12 phases of TSS were formed during barrier progradation, and their mechanism of formation seems to be related to cyclic processes of coastal erosion that have occurred during barrier progradation [20]. Each phase is connected with the destruction of at least one foredune ridge by storm waves. The Curumim barrier belongs to the barrier-lagoon system IV, as referenced in southern Brazil, formed during the Holocene [21]. This barrier-lagoon system corresponds to a high-frequency depositional sequence [22,23]. The modern beach of the Curumim barrier is formed by fine sand, showing dissipative to intermediate morphodynamic stages, and submitted to a microtidal regime with a mean range of only 0.3 m, and a maximum high spring tide of 0.53 m [24].

Consequently, sediment transport and deposition along the open coast at the Curumim barrier is primarily dominated by moderate wave energy with a mean significant wave height (Hs) of 1.25 m [25]. During autumn and winter storms (April–July) the wave height may frequently exceed 2.0 m, and the sea-level can surge up to 1.3 m [26]. Expected extreme events can increase sea-level up to 3.2 m [24]. The climate is subtropical, constantly humid (Cfa), without a dry season, with hot summer [27]. Rainfall ranges from 1600 to 1900 mm and is evenly distributed throughout the year. NE winds predominate along the entire coast but combined with the secondary winds from the S, a resultant drift direction to WNW is produced [28].

*Sea-Level History*

The postglacial sea-level history of the study area extends back approximately 17.5 ka BP when the sea level was approximately 120 to 130 m lower [29]. After that time, the sea level rose at an average rate of 1.2 cm/year and reached the present level for the first time at around 6 ka BP (this work). As mentioned before, there are no reliable data on sea-level behavior during the middle to late Holocene time along this coast. However, sea-level curves for areas to the north indicate that at the culmination of the Postglacial Marine Transgression (PMT) at around 5 ka BP, the sea level was approximately 1–3 m above its present level, after which it slowly fell [4] (Figure 2A). Relict vermetid incrustations records of a coastal area located 200 km to the north of the Curumim barrier indicate a maximum sea level of +2.1 m at around 5 ka BP [30]. To the south (~700 km), in Uruguay, a sea-level envelope indicates a maximum level varying between +3.2 and +2.1 m occurred at around 6 ka BP, followed by an overall fall [31] (Figure 2B). Also, from Uruguay comes the closest record of modern sea-level rise, which indicates a rate of 4.7 mm/years, from 1961 to 2014 [32]. Records of middle-late Holocene punctual sea-level positions obtained in studies along of the coastal plain of the Rio Grande do Sul [6,9], fit the envelope curve of Figure 2A [4].

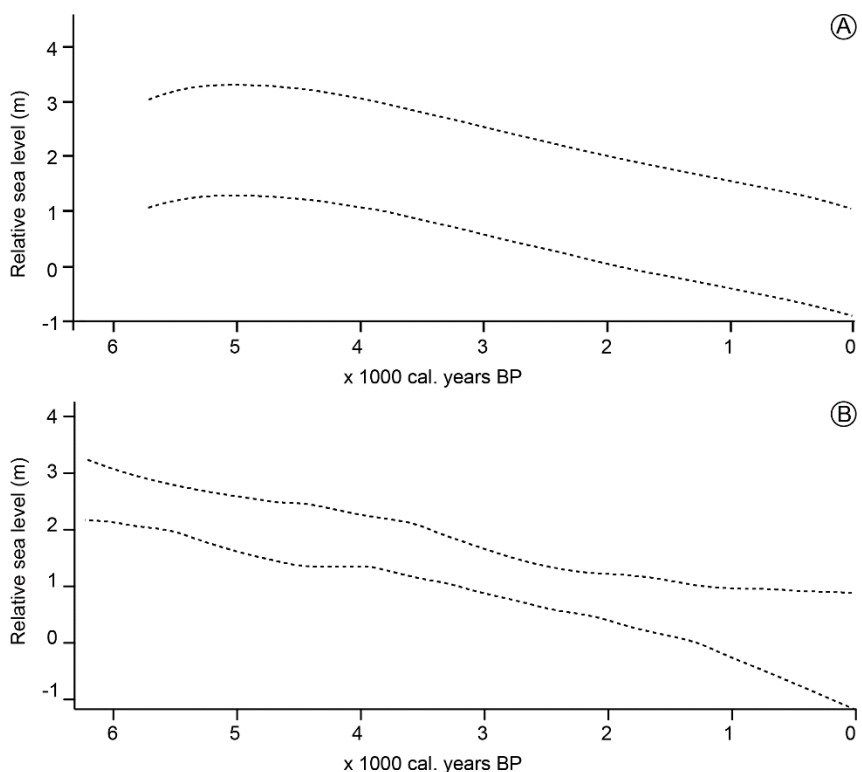

**Figure 2.** (**A**) Sea-level envelope for the Brazilian coast south of 28° S (modified after [4]); (**B**) Sea-level envelope for the Uruguayan coast (modified after [31]).

## 3. Materials and Methods

### 3.1. Ground Penetrating Radar (GPR) Profile

The two-dimensional high-resolution stratigraphic GPR survey was performed on the northern coastal plain of the Rio Grande do Sul State (Figure 1) at the Arroio do Sal (29°30′ S-49°53′ W). The survey was carried out using a four 4WD vehicle, in which the acquisition and positioning units were installed. The utilized GPR system was composed of a GSSI™ (Geophysical Survey Systems, Inc., Nashua, NH, USA) SIR-3000 data collector for monostatic ground-coupled antenna 200 MHz. The GPR profile was collected using the Common Offset method with a two-way-travel time (TWTT) range of 450 ns, which allows for penetration up to 20 m of depth. A vertical IIR low pass (500 MHz), vertical IIR high pass (60 MHz), stacking (3), and gain filters were applied during the time of data acquisition.

The data were post-processed with the Radan™ and Prism2® software packages after background removal and the application of bandpass frequency filters, Ormsby bandpass filter, gain equalization, zero-point, topographic corrections, and time-to-depth conversion. The trace analysis was done, and a dielectric constant of 10 for wet sand was used to convert travel time to depth, which represents a velocity of 0.09 m/ns [33–35]. This constant was validated using lithological data obtained from SPT drill holes [9,36,37].

The GPR profile interpretation was based on the seismostratigraphy method [38] adapted for GPR [39]. The method is based on reflections termination (onlap, downlap, toplap, and truncations), geometry, and pattern of reflections [40–43]. These properties of the reflectors together with their continuity characteristics and signal amplitude were the criteria to define the point of contact between the base of the foredune and the beach (backshore) deposits with better precision. Records of short-term extreme events of sea-level rise are easily identified as erosional surfaces truncating foredune internal reflections.

### 3.2. Global Navigation Satellite System (GNSS) Profile

The GPR profile was topographically corrected using GNSS post-processed elevation data points collected along the profile lines at interval times of 1 s. These data were acquired using a GNSS Trimble® (Sunnyvale, CA, USA) ProXRT (datum: WGS84) and analyzed in a Geographic Information System (GIS). The data were post-processed with the Trimble® Pathfinder® Office software (Sunnyvale, CA, USA), resulting in a horizontal and a vertical precision of 0.1 and 0.3 m, respectively.

### 3.3. OSL Ages

Four samples from aeolian deposits of TSS, with each sample representing one phase of TSS formation, were collected by horizontal insertion of a pipe into the sidewalls cut for road construction, following the classical protocol to avoid any exposure to sunlight. They were dated at the Luminescence Dating Laboratory, Victoria University of Wellington, NZ. OSL ages estimated were obtained on quartz grains, whose concentration followed standard procedures and included wet sieving to isolate the 90–200 μm grain size fraction followed by chemical treatments with hydrogen peroxide (10%) and hydrochloric acid (10%) to eliminate organic compounds and carbonates, respectively [44]. Quartz grains (2.58–2.65 g/cm$^3$) were subsequently isolated from heavy minerals and feldspars using lithium polytungstate solutions. Quartz concentrates were then treated with hydrofluoric acid (40% HF) to eliminate the outer rind of grains affected by alpha radiation and remnant feldspars.

Equivalent doses were determined using the single-aliquot regenerative dose (SAR) protocol applied to multigrain aliquots of quartz [45]. Luminescence measurements were done using a standard Riso TL-DA15 measurement system, equipped with 3 Hoya U340 optical filters (stack thickness ~9 mm) to select the ultraviolet luminescence band centered around 340 nm. Stimulation was done by blue diodes, $\alpha$-irradiations were done on the Riso TL-DA15 $^{90}$Sr,$^{90}$Y β-irradiator, calibrated against the Risø National Laboratory, Denmark, to about 3% accuracy. The palaeodoses were estimated by use of the Single Aliquot Regenerative method (SAR). In the SAR method, a number of aliquots are subjected to a repetitive cycle of irradiation, preheat, and measurement. In the first cycle, the natural luminescence output is measured, in all following cycles an artificial dose is applied. Usually, four or five of these dose points are used to build the luminescence growth curve (β-induced luminescence intensity versus added dose) and bracket the natural luminescence output. This allows interpolation of the equivalent dose (the β-dose equivalent to the palaeodose). In order to correct for potential sensitivity changes from cycle to cycle, a test dose is applied between the cycles, preheated ('cut heat'), and measured. For the samples reported here 16 aliquots were measured, preheat temperature was 260 °C for 10 s, cut heat was 220 °C for 10 s, and measurement time 40 s (which resets the luminescence signal to a negligible residual).

Dose rates were calculated through concentrations of uranium (U), thorium (Th), and potassium (K). The dry, ground, and homogenised soil samples were encapsulated in airtight perspex containers and stored for at least four weeks. This procedure minimizes the loss of the short-lived noble gas $^{222}$Rn and allows $^{226}$Ra to reach equilibrium with its daughters $^{214}$Pb and $^{214}$Bi. The samples were counted using high-resolution gamma spectrometry with a broad energy Ge detector for a minimum time of 24 h. The spectra were analysed using GENIE2000 software, and cosmic dose rates were calculated [46]. The dose rate calculation is based on the activity concentration of the nuclides $^{40}$K, $^{208}$Tl, $^{212}$Pb, $^{228}$Ac, $^{210}$Pb, $^{214}$Bi, $^{214}$Pb, $^{226}$Ra (Table 1).

**Table 1.** Radionuclide, water contents, measured dose rate, equivalent dose, and luminescence ages.

| Sample | Water Content δ [1] | U (μg/g) from $^{234}$Th | U (μg/g) [2] from $^{226}$Ra, $^{214}$Pb, $^{214}$Bi | U (μg/g) from $^{210}$Pb | Th (μg/g)[2] from $^{208}$Tl, $^{212}$Pb, $^{228}$Ac | K (%) | dD/dt (Gy/ka) [#] | D$_e$ (Gy) | OSL-Age (ka) |
|---|---|---|---|---|---|---|---|---|---|
| WLL380 | 1.083 | 0.30 ± 0.09 | 0.45 ± 0.01 | 0.29 ± 0.09 | 1.57 ± 0.03 | 0.62 ± 0.01 | 0.918 ± 0.029 | 6.14 ± 0.92 | 6.69 ± 1.02 |
| WLL381 | 1.161 | 0.35 ± 0.08 | 0.47 ± 0.01 | 0.51 ± 0.09 | 1.38 ± 0.03 | 0.35 ± 0.01 | 0.641 ± 0.031 | 3.84 ± 0.66 | 5.99 ± 1.07 |
| WLL383 | 1.194 | 0.26 ± 0.08 | 0.33 ± 0.01 | 0.37 ± 0.09 | 0.95 ± 0.03 | 0.38 ± 0.01 | 0.593 ± 0.034 | 2.98 ± 0.51 | 5.03 ± 0.91 |
| WLL404 | 1.192 | 0.27 ± 0.05 | 0.31 ± 0.01 | 0.42 ± 0.05 | 0.97 ± 0.02 | 0.32 ± 0.01 | 0.543 ± 0.030 | 2.38 ± 0.55 | 4.38 ± 1.04 |

[1] Ratio wet sample to dry sample weight. Errors assumed 50% of (δ-1). [2] U and Th-content is calculated from the error weighted mean of the isotope equivalent contents. [#] The doserate dD/dt was calculated using the conversion factors of Adamiec and Aitken (1998).

### 3.4. Geostatistic

The first curve of Holocene paleo sea levels for southern Brazil (29°30′ S-49°53′ W) is based on 24 positions, in which the distance from shore (m) and altimetric values relative to the present mean sea-level (MSL) are indicated (Table 2). A modern point at (0, 0) was also included to indicate the present mean sea level. The positions were then modeled linearly using the function "lm" with a 5th-order polynomial employing the argument "poly". The predicted intervals are based on 0.99 level confidence, which represents here the envelope curve. It means a 99% confidence that the following new observation (distance of shore) will fall within this range. The adjusted coefficient of determination $R^2$ was determined with a significance level gamma threshold value of 0.05. All analyses were carried out in the open-source R language [47].

**Table 2.** Table showing the distances of the 24 points identified, with their respective mean and corrected altitudes, associated with the correlated OSL ages.

| Distance (m) | Altitude (m asl) GNSS | 68% Accuracy HZ (m) | 68% Accuracy VT (m) | Altitude of Paleo Sea Levels (m asl) | Correlated OSL Age (ka BP) |
|---|---|---|---|---|---|
| 165.30 | 2.0 | 0.1 | 0.3 | 0.5 | |
| 265.80 | 2.3 | 0.1 | 0.3 | 0.8 | |
| 364.70 | 2.5 | 0.1 | 0.3 | 1.0 | |
| 420.80 | 2.4 | 0.1 | 0.3 | 0.9 | |
| 523.20 | 2.3 | 0.1 | 0.3 | 0.8 | |
| 657.30 | 3.0 | 0.1 | 0.3 | 1.5 | |
| 743.50 | 3.0 | 0.1 | 0.3 | 1.5 | |
| 1000.00 | 3.0 | 0.1 | 0.3 | 1.5 | 4.38 ± 1.04 |
| 1737.26 | 3.2 | 0.1 | 0.3 | 1.7 | |
| 2320.00 | 3.3 | 0.1 | 0.3 | 1.8 | 5.03 ± 0.91 |
| 2411.56 | 3.3 | 0.1 | 0.3 | 1.8 | |
| 2457.56 | 3.3 | 0.1 | 0.3 | 1.8 | |
| 2557.16 | 3.4 | 0.1 | 0.3 | 1.9 | |
| 2596.16 | 3.3 | 0.1 | 0.3 | 1.8 | |
| 2733.93 | 3.3 | 0.1 | 0.3 | 1.8 | |
| 2988.93 | 3.2 | 0.1 | 0.3 | 1.7 | |
| 2997.83 | 3.1 | 0.1 | 0.3 | 1.6 | |
| 3070.63 | 2.4 | 0.1 | 0.3 | 0.9 | |

**Table 2.** *Cont.*

| Distance (m) | Altitude (m asl) GNSS | 68% Accuracy HZ (m) | 68% Accuracy VT (m) | Altitude of Paleo Sea Levels (m asl) | Correlated OSL Age (ka BP) |
|---|---|---|---|---|---|
| 3301.43 | 2.2 | 0.1 | 0.3 | 0.7 | |
| 3458.83 | 1.9 | 0.1 | 0.3 | 0.4 | |
| 3580.00 | 1.0 | 0.1 | 0.3 | −0.5 | 5.99 ± 1.07 |
| 3705.83 | −0.2 | 0.1 | 0.3 | −1.7 | |
| 3917.53 | −0.4 | 0.1 | 0.3 | −1.9 | |
| 3967.73 | −0.7 | 0.1 | 0.3 | −2.2 | |
| 3975.00 | −1.7 | 0.1 | 0.3 | −3.2 | 6.69 ± 1.02 |
| 3986.63 | −1.9 | 0.1 | 0.3 | −3.4 | |
| 4019.63 | −2.2 | 0.1 | 0.3 | −3.7 | |
| 4041.03 | −2.6 | 0.1 | 0.3 | −4.1 | |

## 4. Results

The analysis and interpretation of the GPR record obtained along a 4.3 km long profile made possible the identification of different patterns of reflectors, which were separated as distinct radarfacies (Rdf) (Figure 3).

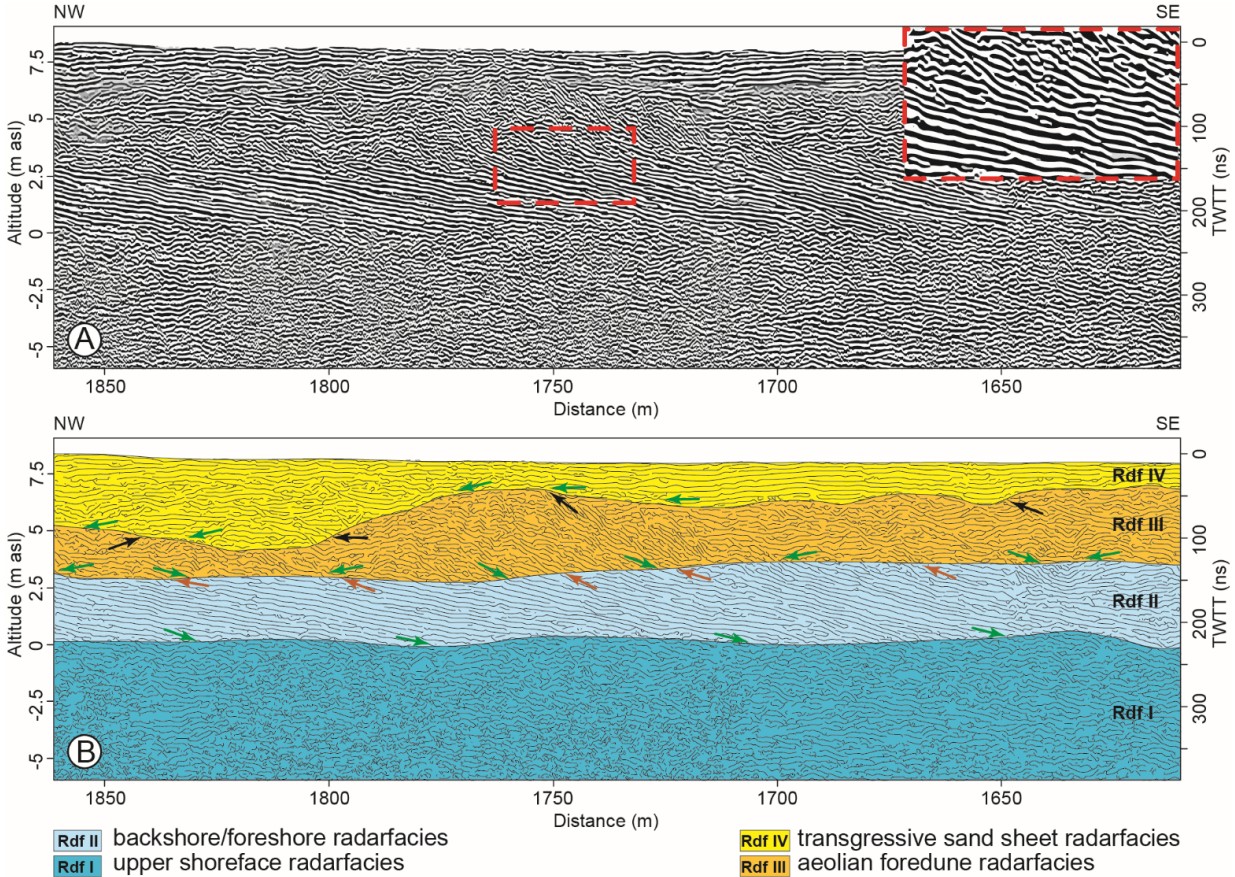

**Figure 3.** The short interval of the GPR profile above was performed at the Arroio do Sal (see location in Figure 1). (**A**) Processed section showing, in the dashed red rectangle, the contact between the foredune and the backshore (zoom in for a better observation); (**B**) Interpreted section showing the four radarfacies. The arrows represent the reflection terminations (toplap: orange, downlap: green, truncations: black). See insertion of this GPR interval in the large and stacked GPR record of Figure 5B.

Rdf I presents irregular, non-continuous, and sub-parallel reflectors in a general segmented undulating pattern, commonly found at the upper shoreface. Rdf II shows a downlap termination over radarfacies I and a toplap termination with radarfacies III. The downlap and toplap terminations characterize an oblique-parallel progradational configuration (clinoforms) formed under a high-energy depositional condition, typical of the backshore/foreshore zone. Rdf III shows reflectors in downlap termination over the clinoforms of Rdf II for both directions (landwards and seawards), a marked characteristic of foredunes. The upper limit of this radarfacies is marked by a continuous and undulating high amplitude reflector. Rdf V has continuous, sub-parallel reflectors with a low-angle downlap termination in the landward direction, corresponding to the aeolian TSS that dominate at the barrier surface [48–50].

As demonstrated by many authors, the contact between foredune and backshore deposits is strongly correlated with sea-level position, e.g., [9,10,12–14,51]. On coasts such as the one of Curumim barrier, the low micro-tidal range (0.3 m) makes wave energy (maximum runup during regular storms) the main factor controlling the vertical distance of this contact in relation to the present sea-level position. On this southern coast of Brazil, this vertical distance ranges from 1.0 to 1.5 m [9]. At the Curumim barrier, the modern contact between foredune and backshore deposits can be identified by the downlap and toplap terminations of reflectors representing foredune and backshore deposits, respectively (see Figures 3 and 4). Eleven (11) measurements made at the modern (active) dune-beach system of the Curumim barrier in 2003 and 2004 (unpublished data) indicated an average vertical distance of $1.5 \pm 0.4$ m between this contact and the mean sea level. This vertical distance is also found 50 km to the north of the Curumim barrier [8]. In contrast, on coasts such as Troia Peninsula (Portugal) (macro-tidal range) and at The Granites (South Australia) (moderate wave energy but macro-tidal range), this vertical distance is around 4.6 and 3.0 m, respectively [13,52]. Thus, it is clear the great influence of the tidal range on the altitude of the contact between foredune and backshore deposits. With the assumption that this vertical distance has remained almost the same during the Holocene evolution of prograded barriers (meaning relatively constant wave energy and tidal range), it is possible to access past sea-level positions by subtracting the vertical distance from the heights of the contact of relict foredune and backshore deposits identified in GPR records [53].

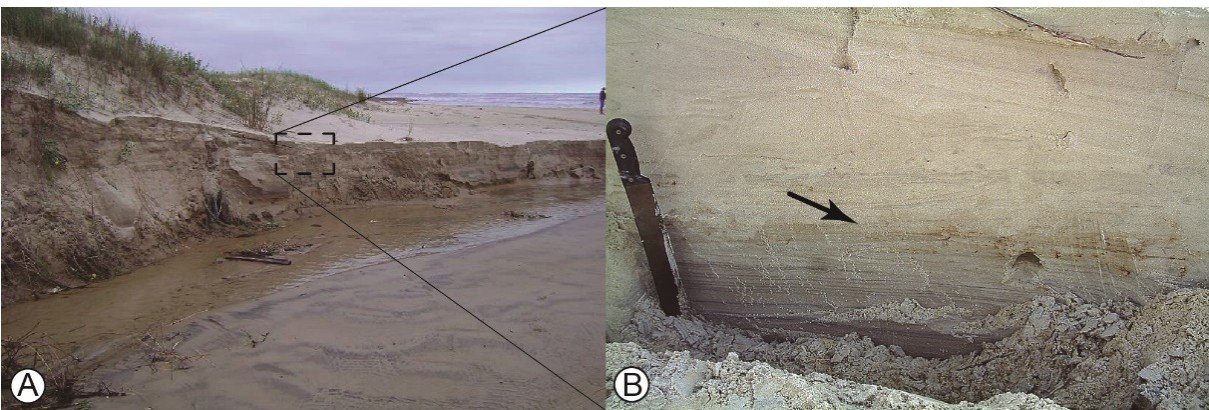

**Figure 4.** (**A**) The morphological transition between the foredune and the backshore at the Curumim barrier and (**B**) detail of the sharp contact (black arrow) between backshore sands (light grey color) and foredune sands (white to pale yellow). At this place the contact is located 1.5 m above mean sea level (picture orientation is NW-SE).

A total of 24 points precisely marking the contact of downlap and toplap terminations between foredune and backshore radarfacies were determined (Figures 3 and 5). Figure 3 presents 250 m only of the 4.3 km long GPR profile performed at Arroio do Sal on the Curumim barrier (Figures 1 and 5). The portions along of the GPR profile in Figure 5, where the contacts between the foredune and the backshore deposits were not identified, are related to places eroded by channelized waters.

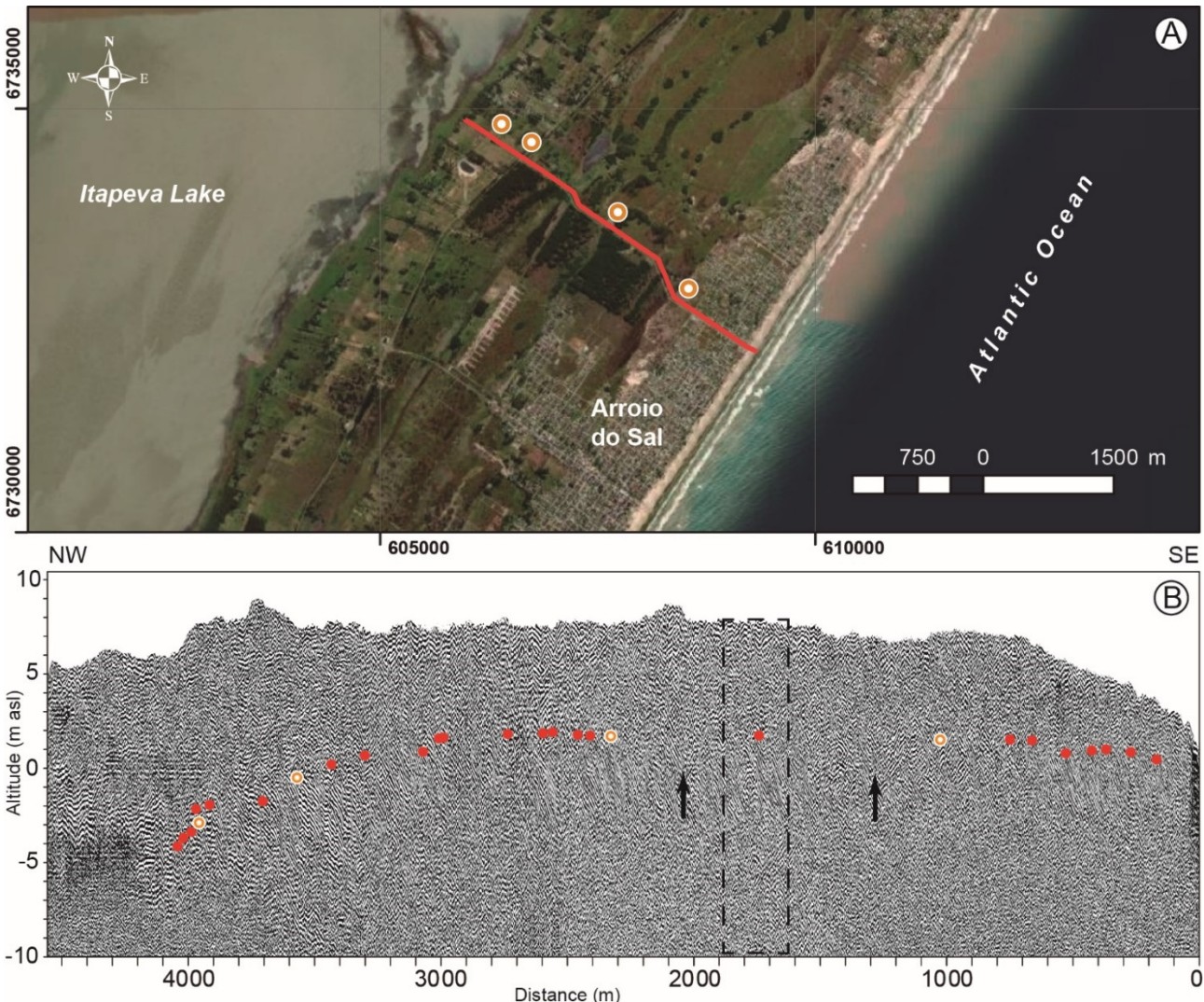

**Figure 5.** (**A**) Location of the GPR profile along the Arroio do Sal (UTM–Datum: WGS84). (**B**) Correspondent GPR section. The 24 red dots indicate the positions of the contacts between the foot of the foredune and the backshore. The large intervals along the profile showing no dots correspond to places eroded by channelized waters (black arrows). The orange points represent a projection of the location of OSL ages. The black dashed rectangle corresponds to the location of the GPR profile of Figure 3. Four enlarged (detailed) GPR sections from where the 24 contacts between the foot of the foredune and the backshore were extracted are presented as Supplementary Material.

Relative ages of these contacts were obtained in the form of four OSL ages (Table 1), whose projections are presented together with the GPR profile in Figure 5. Each analyzed sample provided the age of a starting point of a TSS, which in turn approximately corresponds to the location of a former foredune ridge. The four ages inform the chronology of barrier progradation and consequently inform the relative chronology of the formation of the 24 contacts between foredune and backshore radarfacies presented in Figure 5.

After subtracting the 1.5 m vertical distance (the vertical distance between the modern contact and the present position of sea-level–Figure 4B), 24 paleo sea levels were determined (Table 2).

The fitted sea-level variation curve, presented in Figure 6, shows a promising and statistically significant outcome ($R^2$ = 0.966, *p* < 0.001). The predicated confidence intervals at 99% level (Figure 6; black dashed lines) agree with the envelopes [4,28] presented in Figure 2. It is noteworthy that the curve, notwithstanding be bidimensonally reached, using the distance from shore versus sea-level, encompasses an age-related tridimensional

understanding, including age-estimates prior to the maximum sea-level reached at about 5 ka BP.

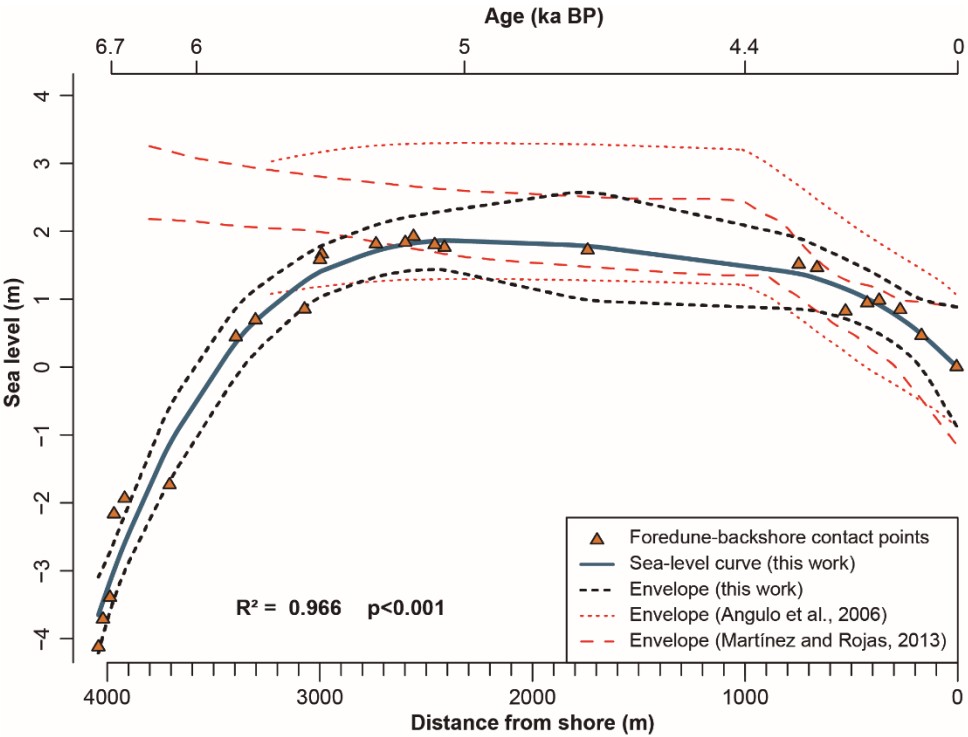

**Figure 6.** The sea-level curve of the southernmost Brazilian coast, this work, based on 24 foredune-backshore contact points identified by GPR at Arroio do Sal (29°30′ S-49°53′ W–Figure 5). These paleo sea levels were adjusted to the current mean sea level (MSL). The 5th-order polynomial curve ($R^2$ = 0.966, $p < 0.001$) and the envelope (black dashed lines), this work, represent the lower and upper limit of the predicted intervals (CI, 99%). OSL ages are displayed at the superior x-axis. We have also included a modern point at (0, 0) to indicate the present mean sea level. The red dotted and dashed lines represent the envelopes of Imbituba-Laguna [4] and Uruguay [31], respectively. These two envelopes were plotted according to axes Y (sea level altitude) and X (age).

## 5. Discussion

The coastal region between the Santa Marta Cape (28°35′ S-48°49′ W–Figure 1) and La Coronilla (33°55′ S-53°30′ W–Figure 1) is 750 km long, representing the emergent portion of the Pelotas oceanic marginal basin, showing essentially Quaternary age deposits. In the last 7 ka, a coastal barrier was formed as a shore parallel geomorphological structure, continuous for 750 km [15]. Immediately to the north of the Santa Marta Cape, at Imbituba-Laguna (28°14′ S-48°39′ W), spatial positions of relict vermetid incrustations and corresponding [14]C datings indicated that +2.1 m was the highest position reached by sea level during middle Holocene (6–5 ka BP), followed after that by an overall sea-level fall [29]. Later, a sea-level envelope was presented for the Imbituba-Laguna coastal region, indicating a maximum sea level varying from +3 to +1 m, reached at around 5 ka BP [4].

The precision of the spatial position of the 24 indicator points of past sea-level, determined in GPR records, is controlled by the operational basis (fundamentals) of the GNSS system and of the GPR system itself. In regard to the GPR system, precision is basically related to the velocity and frequency of the electrical signal in the unconsolidated sediments (sand-dominated in the case of this study). However, as the depths of the target contacts are less than 5 m and considering that contacts were also determined directly in 14 cores obtained from drill-holes [36], a much more real dielectric constant was applied (10, for wet sand), and thus precision seems to be remarkably high. The frequency of the GPR system (200 MHz) corresponds to a reflector resolution of 0.25 to 0.30 m [54,55]. A

precision of 0.1 m (horizontal) and 0.3 m (vertical) was determined regarding the GNSS system. Thus, the altitudes of the target points are subjected to a maximum error of ±0.6 m. Indeed, the less precision value used to build the proposed sea-level curve is the vertical distance of 1.5 m, subtracted from the altitudes of the GPR reflectors representing the 24 relict contacts between foredune and backshore deposits identified along the prograded plain of the Curumim barrier. As mentioned above, this 1.5 m value was obtained from average measurements of 11 surveys performed at the modern dune-beach system of the Curumim barrier in 2003 and 2004, and from a survey made 50 km to the north of the Curumim barrier [8]. Moreover, as this vertical altitude is mainly controlled by wave energy acting on the Curumim barrier, due to the small average tidal range of 0.3 m, it was assumed that wave energy and consequently this average value of 1.5 m were kept constant during 7 ka of barrier progradation.

An exercise of validation of the sea-level curve obtained from the Curumim barrier was done by comparison with nearby data of sea-level changes during the Holocene. At Imbituba-Laguna (160 km to the north of Arroio do Sal), the maximum sea level of +2.1 m, around 5 ka BP [30] was obtained from fossil vermetid records whose precision is indicated by a standard deviation of ±1 m. This standard deviation results from variations of the altitude of the vermetid bioconstructions produced by variations of wave energy and tidal range [56,57]. The variations also result from the morphology of the rocky coast, the homologous reference level considered, and the preserved part of the bioconstruction [58].

Table 2 shows that at Arroio do Sal, between 5.99 and 5.03 ka BP, altitudes of sea-level positions varied between 1.9 and 0.4 m. Thus, independently of the margin of errors of the two methods of paleo sea-level determination, the maximum sea levels that occurred at Arroio do Sal (1.9 m) and at Imbituba-Laguna (2.1 m), at around 5 ka BP, are remarkably similar. To the south of Arroio do Sal, the nearest sea-level data are distant by approximately 700 km, embracing a large coastal area of the Atlantic Ocean (Argentina and Uruguay) and the La Plata River [31,59,60]. Those data are mainly based on sea-level indicators extracted from beach ridges and beach storm deposits, and they have been intensely debated in regard of their meaning as sea-level indicators for the last 7 ka of this coastal region [31,59,60]. One of the above studies interpreted a maximum level of +4 m at around 6 ka BP and two high-frequency oscillations of sea level that may have occurred at around 4.3 and 2.0 ka BP [59], while the other two studies interpreted a continuous sea-level curve, progressively falling to the present position, however, they disagree between each other by around 1 m in regard of the maximum level reached at around 6 ka BP [31,60]. Thus, despite the disagreement between researchers studying sea-level behavior at Argentina and Uruguay, some general similarities with the sea-level record obtained at Arroio do Sal could be highlighted, such as the maximum level of few meters reached around 6–5 ka BP and followed by an overall sea-level fall to the present position (not considering high-frequency oscillations). However, when comparing the Arroio do Sal sea-level data with results of Uruguay [31] only, similarities are worthy of note. For the Uruguayan coast a maximum sea level varying between +3.2 to +2.1 m (average +2.65 m), at around 6 ka BP, followed by a continuous and progressively fall [31].

This paper does not discuss the drivers of sea-level behavior in the last 7 ka. However, considering that the Curumim barrier is located in the central part of a marginal sedimentary basin (Pelotas Basin) and that, for instance, the Atlantic coast of Uruguay occurs at a structural high that corresponds to the southern limit of Pelotas Basin, some geological factors such as the differential compaction of sedimentary deposits, and tectonics could come to explain the difference between the maximum sea-levels reached at around 6–5 ka BP, recorded at Arroio do Sal and at Argentina and Uruguayan coasts. For instance, regarding tectonics is worth of mention that in a south direction, the eastern continental margin of South America gets closer to its counterpart tectonic-active western continental margin. These factors have not already deserved appropriate investigation by coastal researchers from Argentina, Uruguay, and Brazil. As a second hypothesis, the difference between the maximum sea levels reached at both regions (Uruguay and South

Brazil) around 6–5 ka BP, could be related to the adjustment (correction) of the sea level indicators used to infer the paleo-sea levels at the Uruguayan coast. The altitude of the paleo-sea levels was determined by the subtraction of 1.5 m (considered the storm mean altitude), but the authors informed that the extreme storms recorded in the last 100 years reached 4.7 m [31]. As extreme events commonly produce records with a high potential of preservation, it is possible that the dated storm deposits correspond to storms that reached altitudes higher than 1.5 m, and therefore the inferred altitudes could be over-estimated.

Important to mention here that these extreme storm events would have reached southern Brazil as well. It was mentioned above an expected storm surge of up to 3.2 m [24]. But differently from the paleo sea-level indicators of the Argentine and Uruguayan coast, which are mostly beach storm deposits, the indicator analysed in the progradational record of the Curumim barrier was the depositional contact between the base of the foredune and the backshore deposits, which in turn was formed under a combination of fair-weather conditions and regular (not extreme) storms. Extreme storm events are recorded in GPR profiles, but they are easily identified as erosional surfaces truncating the foredune internal reflections.

The curve extracted from the Curumim barrier shows that sea-level was still rising before 6 ka BP, with a maximum level of 1.9 m reached close to 5 ka BP; after that, sea-level started to fall slowly until around 4 ka BP when fall was accelerated. It is not the goal of this paper to discuss the geological consequences of the sea-level behavior indicated by the curve of the Curumim barrier; however, it is important to raise the attention to the possibility that the fall acceleration after 4 ka BP may have resulted in changes over coastal systems of southern Brazil. For instance, palynology studies of Holocene lagoonal deposits of southern Brazil (28°01′ S-48°38′ W and 33°43′ S-53°21′ W) [61,62], respectively, found that the lagoonal systems have changed to a freshwater dominance after 4–3 ka BP. Other potential consequences deserve future investigation.

The global sea-level rise driven by global warming was not detected in the sedimentary records of the prograded barriers of the Pelotas Basin. Historical aerial pho-tographies of the last 40 years indicated that the Curumim barrier is still prograding at a rate of 3.0 to 3.4 m/year [63]. This means that a positive sediment balance may be overcoming the ongoing contemporaneous sea-level rise. As an exercise, by taking into account the rate of sea-level rise of 4.7 mm/year acting for 100 years, an elevation of 47 cm was expected of the contact between foredune and backshore radarfacies, at the GPR records close to the shoreline. The GPR record is blind along the last 150 m due to the saltwater incursion, which made it impossible to verify this probable elevation of the contact between the radarfacies.

At the end of the last century, high-resolution seismic reflection equipment was still used on Quaternary terrains aiming the obtention of continuous stratigraphic records, but at an extremely high cost and too much hard work. Fortunately, after the establishment of the GPR as a tool to investigate geological problems, following the early background fundamentals [64,65], the GPR devices quickly turned to portable and precise operating systems, furnishing conditions to run relatively low-cost quick surveys on Quaternary coastal terrains. Thus, it is gratefully and worthy of mention that after 30 years of pursuing the production of a sea-level curve telling the sea-level history for the last 7 ka of the southernmost coast of Brazil, this goal was finally reached here, although perfectible.

## 6. Conclusions

The two-dimensional GPR profile performed on the prograded barrier of Curumim revealed a detailed stratigraphy, showing patterns of reflectors that allowed the identification of the geometries of foredunes and beach (backshore) deposits. The downlap and toplap reflectors termination of these two deposits allowed to define with high precision their contacts, which in turn have a strong correlation with sea level.

With great confidence, a total of 24 points representing these contacts were identified in the GPR profile, allowing the construction of the first sea-level curve that represents the

changes on sea-level during the last 7 ka, which occurred at the southernmost sector of the Brazilian coast.

The curve shows that sea level was still rising before 6 ka BP, with a maximum level of 1.9 m reached close to 5 ka BP; after that, sea level started to fall slowly until around 4 ka BP when fall was accelerated.

By providing the precise and continuous visualization of the well-defined contact between foredune and backshore deposits on prograded barriers, the GPR system, when combined with high-precision spatial positioning systems and with geochronological data such as OSL, made possible the construction of a curve remarkably similar to curves constructed, for instance, using biological indicators such as fossil vermetid, which have allowed the production of sea-level curves for most of the Brazilian coast in the last 30 years.

**Supplementary Materials:** The following are available online at https://www.mdpi.com/article/10.3390/geosciences11080326/s1, Figures S1–S4: GPR Section 1–4.

**Author Contributions:** Conceptualization, E.G.B. and S.R.D.; investigation, E.G.B., S.R.D., M.L.C.d.C.R. and A.B.d.S.; data processing, E.G.B., M.L.C.d.C.R., M.d.N.R., F.C.; writing—original draft preparation, E.G.B., S.R.D., M.d.N.R. and R.J.A.; writing—review and editing, E.G.B., S.R.D., M.C.d.S., R.J.A., M.L.C.d.C.R., A.B.d.S., F.C. and M.d.N.R. All authors have read and agreed to the published version of the manuscript.

**Funding:** This research was funded by MCTI/CNPQ/Universal 14/2014, grant number 446060/2014-3.

**Institutional Review Board Statement:** The study was approved by the Institutional Review Board of Geoscience Institute of UFRGS (Project No.: 29099).

**Informed Consent Statement:** Not applicable.

**Data Availability Statement:** The study did not report any organized data sets.

**Acknowledgments:** E.G.B., S.R.D., R.J.A. and M.L.C.d.C.R. thank the Conselho Nacional de Desenvolvimento Científico e Tecnológico (CNPq) for the provision of their research fellowships.

**Conflicts of Interest:** The authors declare no conflict of interest.

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
