# Peer review of "Holocene Sea-Level Changes in Southern Brazil Based on High-Resolution Radar Stratigraphy"

_geosciences, doi:10.3390/geosciences11080326_

Round 1

Reviewer 1 Report

The paper entitled “Holocene Sea-level Changes in Southern Brazil (29°30’S) Based on High-resolution Radar Stratigraphy” is of a very interesting topic absolutely suitable for the journal “Geosciences”. I am not a native speaker but I think that the manuscript reads well.

The methodology is generally well-explained and the results are well-presented. The conclusions are based on the results but the authors should expand the discussion regarding the main reasons for the observed mean sea-level changes detected in the area.

Here are some comments that could improve the final version of the manuscript:

It is not clear to me how the samples for OSL dating were collected? The sample that was below sea-level was collected through drilling? The other samples were collected from a subaerial low cliff where the stratigraphic layers are visible?

Regarding the OSL dating I think that a very short section describing the basic steps of the methodology and a short paragraph describing the results of the dating is necessary.

I was wondering about the tectonic setting of the region. Since I am not familiar with the study area I would appreciate if the authors add a paragraph at the “Regional Setting” section bout the tectonic status of the area. Is the area stable? Is it affected by any fault?

An explanation about the sea-level fluctuation should be given at the discussion. What is the main reason for sea-level changes? The eustatic component? Is there any influence by the tectonic activity of the area? If yes in what way does tectonic activity affects the relative sea-level change? I think that a more detailed discussion about the sea-level changes should be done taking into consideration both sediment compaction and the local tectonic setting of the study area. Of particular interest is the explanation of the sea-level fall. What the main reason is since as we all know mean sea-level is rising globally.

Regarding the sediment compaction the authors should provide an idea about the mean subsidence rate of the barrier surface due to the long term compaction of the sediments.

Author Response

Reviewer # 1: Many thanks for your comments/suggestions.

Reviewer:

It is not clear to me how the samples for OSL dating were collected? The sample that was below sea-level was collected through drilling? The other samples were collected from a subaerial low cliff where the stratigraphic layers are visible?

Regarding the OSL dating I think that a very short section describing the basic steps of the methodology and a short paragraph describing the results of the dating is necessary.

Our Answer: at section 3.3. OSL Ages we added full information of field and lab procedures, including a new Table (Table 1). Also, in the Results section (second paragraph after Figure 4), you will find the following explanation:

“Relative ages of these contacts were obtained in the form of four OSL ages (Table 1). Each analyzed sample provided the age of a starting point of a TSS, which in turn approximately corresponds to the location of a former foredune ridge. The four ages inform the chronology of barrier progradation and consequently inform the relative chronology of the formation of the 24 contacts between foredune and backshore radarfacies presented in Figure 5.”

Reviewer:

I was wondering about the tectonic setting of the region. Since I am not familiar with the study area I would appreciate if the authors add a paragraph at the “Regional Setting” section bout the tectonic status of the area. Is the area stable? Is it affected by any fault?

Our answer: in section 2. Regional Setting (first paragraph) we added information about the tectonic status, including the present general understanding about basin subsidence and sediment compaction during the Holocene.

Reviewer:

An explanation about the sea-level fluctuation should be given at the discussion. What is the main reason for sea-level changes? The eustatic component? Is there any influence by the tectonic activity of the area? If yes in what way does tectonic activity affects the relative sea-level change? I think that a more detailed discussion about the sea-level changes should be done taking into consideration both sediment compaction and the local tectonic setting of the study area. Of particular interest is the explanation of the sea-level fall. What the main reason is since as we all know mean sea-level is rising globally.

Our answer: in the fourth paragraph, after Figure 6 we started saying that this MS does not discuss drivers of sea-level behavior in the last 7 ka, and differences of the maximum sea level during the Holocene are discussed in theory (due to the lack of specific information). There are a lot of speculations in regard to the overall sea-level fall after 6-5 ka (geoidal deformation, continental uplift, hydroisostasy, etc). The fact is that most of the southern hemisphere presents such behavior.

The paragraph below was included at the end of the Discussion section as a tentative to explain why there is no record of the anthropogenic caused contemporaneous sea-level rise.

“The global sea-level rise driven by global warming was not detected in the sedimentary records of the prograded barriers of the Pelotas Basin. Historical aerial photographs of the last 40 years indicated that Curumim barrier is still prograding at a rate of 3.0 to 3.4 m/year [63]. This means that a positive sediment balance may be overcoming the ongoing contemporaneous sea-level rise. As an exercise, by taking into account the rate of sea-level rise of 4.7 mm/year acting for 100 years, an elevation of 47 cm was expected of the contact between foredune and backshore radarfacies, at the GPR records close to the shoreline. The GPR record is blind along the last 150 m due to the saltwater incursion, which made it impossible to verify this probably elevation of the contact between the radarfacies.”

Reviewer:

Regarding the sediment compaction the authors should provide an idea about the mean subsidence rate of the barrier surface due to the long term compaction of the sediments.

Our answer: unfortunately, this information does not exist. As mentioned above, Contreras et al. (2020) indicated that basin subsidence and sediment compaction were not relevant during the Holocene. Ok, maybe not relevant could mean 1, 2, or 3 m (?), which would be significant to this study, but unfortunately, this information was not produced yet. Therefore, we presented a theoretical discussion only.

Reviewer 2 Report

The manuscript describes the local coastal stratigraphy of a southern Brazil transition environment through GPR surveys, 14C, and OSL datings, aiming to reconstruct the Holocene sea-level variations of the Curumin prograded sandy barrier.

The manuscript is well organized, figures and tables are sufficient and almost clear, Materials and Methods, Results, Discussion, and Conclusions sections are quite congruent.

In the text, about 20 articles of Dillenburg S.R. and Barboza E.G., coauthors of this manuscript, are cited. Among these articles, in the References there are some of them very similar both in English and Portuguese: I suggest reducing these self-citations preserving the most significant ones.

Some typos, comments, and suggestions are hereinafter listed.

TEXT

Title, L2: (29°30’S) Why only the latitude? Geographic coordinates are two, so the longitude is missing. In the title this is superfluous, delete it.

Abstract, L19, and L42, L46, L53, L133, L168, L258, L269: 29°30’S and other latitudes. See comment above. In these cases, add the longitude or delete the latitude.

L24, L49, L263, L265, L275, L314, L384, L385: "sea-level" is sea level

L25-26, L35, L50, L56, L112-113, L115, L118, L120-121, etc.: "6 ka ... 5 ka ... 4 ka ... 6-5 ka ... 17.5 ka ... " I know that depending on the dating technique, sometimes it is specified others not: here, do you mean BP? Check it in all the text.

L26: " was accelerated." accelerated.

L42 and L54: "South Brazil", "south Brasil". Uniform it.

L43-44: "paleo sea-level" is paleo sea level

L49: "+3 m" asl?

L70: "their" these

L82, 2. Regional Setting: Is/was this area affected or not by tectonics or subsidence? Anyway, briefly specify it to help readers better understanding the geological outline of the study area.

L119, and L262: "Vermetid", "relict vermetid" incrustations

L125: And currently, what is the relative/absolute sea-level situation? Is it falling or rising? I suggest adding a brief sentence, here.

L149: "signal's amplitude" is "signal amplitude"

L158: "GPS" is for Global Positioning System, specify in brackets as you did for GPR, OSL, GIS, TSS.

L168: "paleo-sea-levels " paleo sea levels

L182: "Rdf" means radarfacies (radar facies)?

L214-L251: "With the assumption ...  in GPR records", I am not sure that wave energy was almost constant during the entire Holocene: actually, during the climatic optimum (at least two), the temperature variations could have increased sea storms and wave energy, unlike the tides (dependent on astronomical forcings). I suggest adding at least a citation here, to reinforce your statement, or change this sentence. 

L293-295: "it was assumed ... constant during 7 ka of barrier progradation." See comment above to L214-251.

L299: "from vermetid" from fossil vermetid

L301-302: "by variations of wave energy and tidal range" No tectonics nor subsidence? See comment to L82 (2. Regional setting) and clarify.

L326-335: These statements about tectonics and neotectonics (a term no more used) should be anticipated (or moved) to the 2. Regional setting section. Introducing these explanations here in the Discussion does not unravel the readers' previous doubts.

L339 and L340: "paleo-sea" paleo sea

L361: " palynology studies", These should be considered with caution.

L373: " this goal was finally reached here" although perfectible.

L391: "vermetid", fossil vermetid

FIGURES

Figure 1, caption: Specify the geographic coordinate system, e.g., WGS84 or other. In the legend, "Aluvial" is Alluvial, correct it.

Figure 2A, B: "Relative sea-level" of Y-axis is Relative sea level.

Figure 3 and its caption, L195-196: " (zoom in for a better 195
observation).;" Delete the dot or semicolon. It would be better to add a zoomed window of this stratigraphic contact in the figure, e.g., above right. Y-axis, left: Altitude (m), add asl in brackets; Y-axis, right: TWT, do you mean TWTT (see L137)? If yes, correct it.

Figure 4, caption: at the end, add the picture orientation, e.g., S-N view. This helps readers a lot.

Figure 5A: The geographic coordinates along the frame (labeling) are kilometric, while in Fig.1 they are in degrees (°). Use the same representation in both figures. Figure 5B: Y-axis "Altitude (m)" add asl in brackets (see comment to Figure 3).

Figure 6: "Sea-level" of the Y-axis is Sea level. "Age (ka)" on X-axis above, is it (ka BP)? Caption, L272: "line", Delete it.

TABLES

Table 1, 2nd column:" Altitude (m) GNSS" Is "m asl"? If yes (e.g., see L223), add it; 5th column: "Altitude of paleo sea levels" Do you mean (m asl)? If yes, add it; 6th column: " Correlated OSL age [17]" (ka BP) is missing: check it.

Author Response

Reviewer # 2: Many thanks for your comments/suggestions.

Reviewer:

In the text, about 20 articles of Dillenburg S.R. and Barboza E.G., coauthors of this manuscript, are cited. Among these articles, in the References there are some of them very similar both in English and Portuguese: I suggest reducing these self-citations preserving the most significant ones.

Our answer: Yes. We agree. We took out the following references:

[17] Dillenburg, S.R.; Barboza, E.G.; Rosa, M.L.C.C.; Caron, F.; Guadagnin, F.; Hesp. P.A. The complex progradational plain of the Curumim barrier in southern Brazil: an archive of cyclic aeolian activity and sea-level changes in the Holocene. (in preparation). 2021.

[31] Dillenburg, S.R.; Barboza, E.G.; Tomazelli, L.J.; Hesp, P.A.; Clerot, L.C.P.; Ayup-Zouain, R.N. The Holocene Coastal Barriers of Rio Grande do Sul. In Geology and Geomorphology of Holocene Coastal Barriers of Brazil; Dillenburg, S.R., Hesp, P.A., Eds.; Springer, Switzerland, 2009; Lecture Notes in Earth Sciences 107, pp. 53–91. https://doi.org/10.1007/978-3-540-44771-9_3.

[32] Barboza, E.G.; Rosa, M.L.C.C.; Caron, F. Metodologia de Aquisição e Processamento em Dados de Georradar (GPR) nos Depósitos Quaternários da Porção Emersa da Bacia de Pelotas. In: VI Simpósio Brasileiro de Geofísica. Resumos Expandidos 1, 2014; 1–6. https://doi.org/10.13140/2.1.3369.5047

[36] Dillenburg, S.R.; Barboza, E.G.; Hesp, P.A.; Rosa, M.L.C.C. Ground Penetrating Radar (GPR) and Standard Penetration Test (SPT) records of a regressive barrier in southern Brazil. J. Coast. Res. 2011, SI 64, 651–655.

[43] Barboza, E.G.; Rosa, M.L.C.C.; Hesp, P.A.; Dillenburg, S.R.; Tomazelli, L.J.; Ayup-Zouain, R.N. Evolution of the Holocene Coastal Barrier of Pelotas Basin (Southern Brazil) - a new approach with GPR data. J. Coast. Res. 2011, SI 64, 646–650.

Reviewer:

 Title, L2: (29°30’S) Why only the latitude? Geographic coordinates are two, so the longitude is missing. In the title this is superfluous, delete it.

Abstract, L19, and L42, L46, L53, L133, L168, L258, L269: 29°30’S and

other latitudes. See comment above. In these cases, add the longitude or delete the latitude.

L24, L49, L263, L265, L275, L314, L384, L385: "sea-level" is sea level

L25-26, L35, L50, L56, L112-113, L115, L118, L120-121, etc.: "6 ka ... 5

ka ... 4 ka ... 6-5 ka ... 17.5 ka ... " I know that depending on the dating technique, sometimes it is specified others not: here, do you mean BP? Check it in all the text.

L26: " was accelerated." accelerated.

L42 and L54: "South Brazil", "south Brasil". Uniform it.

L43-44: "paleo sea-level" is paleo sea level

L49: "+3 m" asl? L70: "their" these

Our answer: all the above-indicated corrections were done.

Reviewer:

L82, 2. Regional Setting: Is/was this area affected or not by tectonics or subsidence? Anyway, briefly specify it to help readers better understanding the geological outline of the study area.

Our answer: in section 2. Regional Setting (first paragraph) we added information about the tectonic status, including the present general understanding about basin subsidence and sediment compaction during the Holocene.

Reviewer:

L119, and L262: "Vermetid", "relict vermetid" incrustations

Our answer: done

Reviewer:

L125: And currently, what is the relative/absolute sea-level situation? Is it falling or rising? I suggest adding a brief sentence, here.

Our answer: at section 2.1. Sea-level History we added the sentence below:

 “Also from Uruguay comes the closest record of modern sea-level rise, which indicates a rate of 4.7 mm/years, from 1961 to 2014 [32].”

Reviewer:

L149: "signal's amplitude" is "signal amplitude"

L158: "GPS" is for Global Positioning System, specify in brackets as you did for GPR, OSL, GIS, TSS.

L168: "paleo-sea-levels " paleo sea levels

Our answer: done

Reviewer:

L214-L251: "With the assumption ... in GPR records", I am not sure that wave energy was almost constant during the entire Holocene: actually, during the climatic optimum (at least two), the temperature variations could have increased sea storms and wave energy, unlike the tides (dependent on astronomical forcings). I suggest adding at least a citation here, to reinforce your statement, or change this sentence.

L293-295: "it was assumed ... constant during 7 ka of barrier progradation." See comment above to L214-251.

Our answer: You are right, wave energy certainly varied during the Holocene. As we have information of the last years or few decades only, this is a critical assumption. However, similar curves of sea-level produced by Costas et al. 2016 (here number [13]), and Dougherty et al. 2019 (here number [53], included to reinforce our statement) were successfully compared with curves obtained by other methods.

Reviewer:

L299: "from vermetid" from fossil vermetid

Our answer: done

Reviewer:

L301-302: "by variations of wave energy and tidal range" No tectonics nor subsidence? See comment to L82 (2. Regional setting) and clarify.

L326-335: These statements about tectonics and neotectonics (a term no more used) should be anticipated (or moved) to the 2. Regional setting section. Introducing these explanations here in the Discussion does not unravel the readers' previous doubts.

Our answer: in section 2. Regional Setting (first paragraph) we added information about the tectonic status, including the present general understanding about basin subsidence and sediment compaction during the Holocene. Unfortunately, detailed or specific information in regard to basin subsidence and sediment compaction are not available for the Holocene (last 7 ka). This is way we presented hypotheses only (in the Discussion section), to explain the regional small differences of the maximum sea-level reached during the Holocene.

Reviewer:

L339 and L340: "paleo-sea" paleo sea

Our answer: done

Reviewer:

L361: " palynology studies", These should be considered with caution

Our answer: it was presented as an example only of changes on coastal systems due to the overall sea-level fall after 6-5 ka.

Reviewer:

L373: " this goal was finally reached here" although perfectible.

L391: "vermetid", fossil vermetid

Our answer: done

Reviewer:

Figure 1, caption: Specify the geographic coordinate system, e.g., WGS84 or other. In the legend, "Aluvial" is Alluvial, correct it.

Figure 2A, B: "Relative sea-level" of Y-axis is Relative sea level.

Figure 3 and its caption, L195-196: " (zoom in for a better 195 observation).;" Delete the dot or semicolon. It would be better to add a zoomed window of this stratigraphic contact in the figure, e.g., above right. Y-axis, left: Altitude (m), add asl in brackets; Y-axis, right: TWT, do you mean TWTT (see L137)? If yes, correct it.

Figure 4, caption: at the end, add the picture orientation, e.g., S-N view. This helps readers a lot.

Our answer: done

Reviewer:

Figure 5A: The geographic coordinates along the frame (labeling) are kilometric, while in Fig.1 they are in degrees (°). Use the same representation in both figures.

Our answer: as a large-scale figure, figure 1 is presented in degrees. Differently, the local scale in UTM of Figure 5A makes it easy for the reader to have the perception of distances.

Reviewer:

Figure 5B: Y-axis "Altitude (m)" add asl in brackets (see comment to Figure 3).

Figure 6: "Sea-level" of the Y-axis is Sea level. "Age (ka)" on X-axis above, is it (ka BP)? Caption, L272: "line", Delete it.

Table 1, 2nd column:" Altitude (m) GNSS" Is "m asl"? If yes (e.g., see L223), add it; 5th column: "Altitude of paleo sea levels" Do you mean (m asl)? If yes, add it; 6th column: " Correlated OSL age [17]" (ka BP) is missing: check it.

Our answer: done

Round 2

Reviewer 1 Report

I think that the final version of the manuscript is improved! Thus, I suggest to be accepted for publication.